

# Wave energy dissipation in the mangrove vegetation off Mumbai, India

Samiksha S. Volvaiker[1], Ponnumony Vethamony[1], Prasad K. Bhaskaran[2], Premanand Pednekar[1], MHamsa Jishad[1#] and Arthur James[3]

[1]Physical Oceanography Division, CSIR-National Institute of Oceanography, Dona Paula, Goa – 403 004, India

[1#] Presently at Space Applications Centre, Ambawadi Vistar, P.O., Ahmedabad – 380015, India

[2]Department of Ocean Engineering and Naval Architecture, Indian Institute of Technology Kharagpur, Kharagpur 721 302, India

[3]Department of Marine Science, Bharathidasan University, Tiruchirappalli

*Correspondence to*: Samiksha S. V. (vsamiksha@nio.org)

**Abstract.** Coastal regions of India are prone to sea level rise, cyclones, storm surges and human induced activities, resulting in flood, erosion and inundation. The primary aim of the study is to estimate wave energy attenuation by mangrove vegetation using SWAN model, and validate the model results with measurements for the Mumbai coastal region. Wave measurements were carried out during 5-8 August 2015 at 3 locations in a transect normal to the coast using surface mounted pressure level sensors in spring tide conditions. The measured data presents wave height attenuation of the order of 52%. The study shows a linear relationship between wave height attenuation and gradual changes in water level in the nearshore region, in phase with the tides. Model set-up and sensitivity analyses were conducted to understand the model performance to vegetation parameters. It was observed that wave attenuation increases with an increase in drag coefficient, vegetation density and stem diameter. For a typical set-up for Mumbai coastal region having vegetation density of 0.175 per $m^2$, stem diameter of 0.3 m and drag coefficient varying from 0.4 to 1.5, the model reproduced attenuation, ranging from 49 to 55%, which matches well with the measured data. Spectral analysis performed for the cases with and without vegetation very clearly portrays energy dissipation in the vegetation area. This study has the potential of improving the quality of wave prediction in vegetation areas, especially during monsoon season and extreme weather events.

## 1. Introduction

Coastal vegetation protects the coast to a certain extent, from the fury of wind waves, storm surge and tsunami. As waves propagate through the vegetation having sufficient width, due to interaction (with roots, stem, and canopy of vegetation), waves lose energy, resulting in reduction in wave height. Though they act as natural buffer along the coastal areas, it is still uncertain as to what extent waves are attenuated by the vegetation. Dalrymple et al. (1984) proposed a formulation for wave damping effects by vegetation, considering vertical extent of cylinders over the water column for normal incident waves at uniform and arbitrary water depths. This study signified the importance of bulk drag coefficient that takes into account all the approximations considered for wave attenuation. This approach has been applied in spectral wave models and calibrated against flume experiment results (Mendez and Losada, 2004; Suzuki et al., 2012; Wu, 2014).





Mazda et al. (1997a, 1997b) and Massel et al. (1999) focused their work on dissipation of wave energy, by introducing bottom friction and vegetation density as extra components of the drag force. Further, Mazda et al. (2006) investigated dissipation of wave energy accounted by thick mangrove foliage during cyclones. It led to the development of a quantitative formulation connecting vegetation characteristics, incident wave conditions

and local water depth. Dalrymple formulation was further extended by Mendez and Losada(2004);they considered drag as the dominant force, and a parametric relation was developed using Keulegan-Carpenter (KC) number, representing wave transformation in a vegetation field. More or less this covers all the physical processes that occur within the vegetation field as it considered density, diameter and vegetation height in the overall estimation of the bulk drag coefficient. SWAN model uses this formulation, which needs calibration of

bulk drag coefficient of particular plant types. The approach of Massel et al. (1999) was further extended by Luong and Massel (2008); they developed a predictive model for wave propagation through a non-uniform forest of changing water depth. It was found that most of the wave energy got dissipated within a short distance of the mangrove forest, and wave attenuation is less in sparse forest compared to denser forest. For the Vietnam coast, Quartel et al. (2007) carried out field experiments and observed that wave attenuation changes with

roughness of the bed (marshy bottom attenuates about four times more than sandy bed). It may be noted that all models consider linear wave theory within the vegetated zone.

Due to the inaccessibility of mangrove forests, a limited number of field studies has been executed in Vietnam, Australia, China and Japan (Brinkman et al., 1997; Mazda et al., 1997a; Mazda et al., 2006; Vo-Luong and Massel, 2006; Quartel et al., 2007; Bao, 2011; Ysebaert et al., 2011; Yang et al., 2011).Different numerical

and analytical models have been proposed in the last few decades in order to reproduce the hydrodynamics within a vegetation field with regard to wave energy dissipation. One of the approaches used is the bottom friction or bed roughness approach (Hasselmann and Collins, 1968; van Rijn, 1989) that accounts for the effect of vegetation in terms of a bottom friction parameter. Bradley and Houser (2009), Mullarney and Henderson (2010), Riffe et al. (2011) and Stratigaki et al. (2011) conducted wave attenuation experiments in the laboratory,

and estimated wave energy dissipation by calculating integral bulk vegetation drag coefficients. Paul and Amos (2011) examined in detail the frequency-based characteristics of wave energy dissipation and drag coefficient in the case of natural vegetation. Other numerical studies related to wave-vegetation interaction include Li and Yan, 2007; Wu et al. 2012; Ma et al., 2013; Zhu and Chen, 2015, 2017; Chakrabarti et al., 2016; Zhao and Chen, 2016; Beudin et al.,2017.

The energy of waves, tides and currents is attenuated via frictional drag introduced by vegetation, and also by bottom friction in shallow water areas. Field measurements conducted elsewhere indicate that higher wave reduction generally occurs when water reaches the leaves of the dense mangrove. The rate of wave reduction also depends on age of trees, species, vegetation density, incoming wave height, thickness of the forest and mangrove forests structures (Bao, 2011; Mazda et al., 2006; Moller, 2006; Quartel et al., 2007; Vo-

Luong and Massel, 2006). Vegetation reaching the water surface and above (i.e., emergent structures) is more effective in reducing wave height than submerged vegetation (Augustin et al., 2009). The mangrove forest off Carter Road, Mumbai (http://www.mangroves.godrej.com/MangrovesinMumbai.htm) is a planted one, and is growing in height for the past 10 years.

Several studies were carried out based on linear wave theory as well as advanced techniques to relate

the characteristics of vegetated beds and their effect on wave attenuation (e.g. Dalrymple et al., 1984; Kobayashi



et al., 1993 and Mendez and Losada, 2004). These studies considered plant specific depth, averaged drag coefficient, vegetation density and stem diameter. In a few large scale studies, 'total friction coefficient' was often applied, and this parameter was varied to achieve the actual wave attenuation.

Very few studies are conducted in the coastal region of India on wave energy dissipation due to vegetation. Narayan et al. (2010) studied the effectiveness of Kanika Sands Mangrove Island near Dhamra in Odisha, India in attenuating cyclone-induced waves using SWAN 40.81 model. Their study showed that the effectiveness is limited by the geometry and distance from the port to the mangrove island. Parvathy et al (2017) investigated the inter-seasonal variability of wind-waves and attenuation characteristics by mangroves in a reversing wind system using a multi-scale nested modelling approach with WAM and SWAN models. This study quantified the relative rate of wave energy dissipation on monthly and seasonal scales as well as spectral energy in the presence of mangroves. Parvathy and Bhaskaran (2017) conducted a sensitivity study with varying bottom slopes on wave attenuation in the presence of mangroves, and their results reveal that the wave height decays exponentially for the mild slope and found to be consistent with the earlier studies, but as the bottom steepness increases, the wave height reduction gradually becomes more. There are patches of mangrove forests along the coast of India, with varying vegetation density and diameter, but most of these areas are inaccessible for deploying sensors and conducting wave measurements. As no measurements were conducted in the coastal region of India to study the wave energy attenuation due to vegetation, an attempt has been made to make measurements off Mumbai (erstwhile, Bombay) and study the effect.

The post-2004 Great Indian Tsunami surveys revealed how vegetation had protected certain regions along the coast of India. Thereafter steps were taken to plant vegetation along specific zones, that are vulnerable to extreme events. Though it may not be possible to make observations during an extreme event, we thought of studying the wave attenuation characteristics in the vegetation zone along the Indian coast atleast during monsoon season, which fairly represent high wave energy condition, as a preliminary initiative to extreme events. In this context, we have chosen Mumbai coastal region, which is an ideal location to study wave energy attenuation due to mangrove forest using modelling and observations. The present study also relates winds, waves and water level, representing high energy scenario. SWAN model was also set-up to simulate the scenarios. The details of measurements, data analysis, estimation of wave energy attenuation and SWAN model set-up with and without vegetation are described in the following sections.

## 2. Study Area

The port city of greater Mumbai along the west coast of India lies between 18°55ʹN and 19°19'N latitude and 72°47ʹE and 73°05ʹE longitude (Fig. 1). The coastline on the west has four major creeks: Manori, Malad, Mahim, and Mahul. All these creeks and tidal inlets have sheltered shores exposed during low tide conditions conducive for the growth of mangroves. The tides are found to be semi-diurnal, with a range of about 3 m during spring tide (Joseph et al., 2009). Coastal currents are primarily driven by tides. During the southwest monsoon, run-off from the rivers and creeks marginally alters the hydrodynamics. The maximum current is about 1.0m/s during spring and 0.5m/s during neap. Vijay et al. (2005) studied the changes in the mangrove habitat around the Mumbai suburban region using remote sensing data.

The total area of mangroves in the Mumbai suburban region has been estimated to be 56.40 km$^2$ (including mud flats) with dense mangroves contributing 45.4% of the total. During 1990 to 2001, a total



mangrove area of 36.54 km$^2$ was lost, indicating a 39.32% decrease in the area of mangroves (Vijay et al., 2005).*Avicennia marina* was found to be the most dominant mangrove species. Rapid developments such as housing, industrialization, coastal reclamation and population density of Mumbai have resulted in degradation of mangroves, except a few areas such as Carter Road, where the mangroves have grown and registered an increase

in height in the last 10 years. Hence, we have conducted field measurements in the coastal region of Carter Road. The study area and measurements carried out are presented in Fig. 2(a&b).

### 3.    Data and Methodology

### 3.1    Mangrove forest in the Carter coastal area, Mumbai

Landsat5 TM (9 January 2015) satellite dataset (Fig. 2c), obtained from the global land cover facility site with a resolution of 30 m, has been used to estimate the distribution of mangroves off Carter Road. This area has been classified based on one of the unsupervised classification approaches, i.e. ISODATA clustering algorithm (Memarsadeghi et al, 2007). The classification is carefully examined using visual analysis, classification accuracy, band correlation and decision boundary. Only five classes sensibly matched the clusters

i.e. water, mudflat, mangrove, vegetation and urban (Fig. 2d). ERDAS 9.1 and ARC GIS 10.1 software have been used to generate the classification maps. As the focus of this study is confined to mangrove region, only the area covering mangroves was calculated, and it is estimated to be about 8 hectares.

### 3.2    Wave measurements

Waves were measured using surface mounted pressure level sensors during 5-8 August 2015 under spring tide conditions in the nearshore region off Carter Road (Fig. 2b). Four sensors (P1, P2, P3 and P4) were deployed in a transect, stretching over a distance (P1-P4) of 70m. The distance between the probes was maintained minimum, because of the limited width of the vegetation. P1 was deployed away from the vegetation area, P2 at a distance of 17m from P1 (just in front of vegetation), P3 at a distance of 35m from P1 and P4

at71m from P1. Wave measurements were continued for one tidal cycle every day during the field campaign. The density of vegetation varied along the transect. The mangroves near P2 were short and not fully grown. The vegetation height was ≈ 2.5m with roots spreading over an area of 1.5sq.m. The height of mangrove vegetation near P3 (≈5m) was higher than those near P2, and also more dense. At the most landward point of the transect (≈70m from P1), the mangroves were more dense and fully grown with an average height of 7m. It was

observed that waves attenuated almost completely before they reached the fourth pressure sensor (P4), and therefore, the observations at P4 are not included. High frequency (8 Hz) pressure measurements were recorded only when the sensors were submerged under the water.

### 3.3    SWAN model set-up for Mumbai coastal region

The third generation numerical model SWAN (Simulating Waves Nearshore) was specifically developed for finite water depth applications (Booij et al., 1999). The governing equation in the model is the wave action balance equation with various source and sink terms. The bathymetry was generated with ETOPO1 Earth Topography (1 minute) data obtained from the National Geophysical Data Centre, USA. Two model domains are considered in this study: an outer domain and an inner domain. The outer domain covers the entire

Indian Ocean from 60°S to 30°N and 15°E to 130°E (Fig. 1a), and accommodates the distant swells propagating





from the South Indian Ocean/Atlantic Ocean into the North Indian Ocean (Aboobacker et al., 2011;Samiksha et al., 2012;Sabique et al., 2012). The outer domain was set for WAVEWATCH III (WW3) model with a spatial resolution of 0.5°×0.5°. More details of WW3 can be found in Tolman (1991) and Tolman et al. (2014). ERA-I winds were used as input to WW3 model. The boundary files containing information on 2D directional wave
spectra extracted from the outer domain were provided as input to the inner domain. The inner domain was set for SWAN with a spatial resolution of 0.01°×0.01° (17°N to 20°N and 70°E to 74°E) (Fig. 1b);as the size of actual patch of the vegetation is approximately of the size of one grid, one single grid was considered with vegetation; ERA-I winds with the resolution of 0.125°×0.125° were used as input.

     The model discretization considers 31 frequency bins ranging from 0.05 to 1.00 Hz on a logarithmic
scale, and 36 directional bins with an angular resolution of 10°. The SWAN setup in the present study used Cavaleri and Malanotte-Rizzoli (1981) wave growth physics, and shallow water triad non-linear interaction using the lumped triad approximation of Eldeberky (1996). The model was initiated with modified white-capping dissipation (Jansen, 1991) which is the default formulation in SWAN model. The quadruplet non-linear wave-wave interaction was computed using the Discrete Interaction Approximation theory (Hasselman et al.,
1985). The depth induced breaking was computed using spectral version of the model with breaking index, $\gamma =$ 0.73 (Battjes and Janssen, 1978). The bottom friction in SWAN was calculated based on the Collins formulation (Collins, 1972) with a friction coefficient, $c_{fw} = 0.02$ m²s⁻³. The model was also run with different bottom friction physics such as MADSEN and JONSWAP available in the model. However, we found that the results are better with Collins formulation. Therefore, all model runs in this study were simulated using Collins bottom
friction.

### 3.4    SWAN model set-up with vegetation

     The best available form to describe the effect of vegetation on wind-waves is representing the vegetation by vertical rigid cylinders, as postulated by Dalrymple et al. (1984). This method provides a
reasonably good physical representation of the vegetation and its implementation in SWAN. The vegetation properties that were considered in this formulation include vegetation height, vegetation diameter, vegetation density and drag coefficient. The calibration parameter, which is important to determine wave dissipation due to vegetation, is the drag coefficient ($C_d$). By varying drag coefficients, different types of vegetation (both stiff and flexible) can be modelled. Burger (2005) first implemented a vegetation module in the SWAN model by
including the most important variables that play a role in the wave attenuation process i.e. vegetation characteristics and hydraulic conditions. Suzuki et al. (2012) further developed this model by including vertical layers such as those seen in mangroves (e.g. bottom layer containing aerial roots, higher layers containing leaves and branches) and horizontal variation in vegetation characteristics (e.g. different species being present in different areas) with angular frequency and wave number in the model. Wave attenuation in vegetation mainly
depends on the geometrical (number of stems, diameter, branching and height) and biophysical (stiffness and buoyancy) characteristics of the vegetation as well as on the hydrodynamic conditions including water depth, wave period and wave height. In our present study, SWAN model was setup to estimate wave height reduction due to actual mangroves, as well as for assumed vegetation by changing the vegetation parameters in the model.

     The calculation of energy loss is based on the actual work carried out by the vegetation due to plant
induced forces acting on the fluid, expressed in terms of Morison equation (Morison et al., 1950):





$$\varepsilon_v = \frac{2}{3\pi} \rho C_d b_v N \left(\frac{gk}{2\sigma}\right)^3 \frac{sinh^3 k\alpha h + 3\ sinhk\alpha h}{3kcosh^3 kh} H^3$$

where, $\varepsilon_v$ is the time-averaged rate of energy dissipation per unit area; $C_d$, $b_v$ and N are the vegetation drag coefficient, diameter and spatial density (number of stands per unit area), $k$ the wave number, $\sigma$ the wave frequency, $\alpha$ the ratio of plant height to water depth, $h$ the water depth and $H$ the wave height at that point. This method neglected vegetation motion such as vibration due to vortices and swaying motions. For relatively stiff plants, the drag forces are dominant and inertial forces are neglected. Moreover, since the drag due to friction is much smaller than the drag due to pressure differences, only the latter is considered (SWAN manual, Cycle III version 41.01A, 2015).

For the vegetation species present in the study region, the control values of vegetation parameters were determined based on literature as well as personal communications with experts in the field. Vegetation height provided in the model considers the average height (3 m); canopy of the mangroves usually remained above MHWL. On an average, the stem diameter of the plants is around 0.3 m. The estimated area of vegetation is around 8 hectares (= 80,000 $m^2$), and the number of mangrove plants estimated from the satellite imagery is 14000. This provided a vegetation density (number of stems/area of vegetation) of 0.175/$m^2$. However, we have conducted numerical experiments by varying the stem diameter from 0.3 m to 0.2 and 0.1 m, and density from 0.20 to 0.35/$m^2$. The sensitivity analyses were carried out by varying the drag coefficients, density of the vegetation and stem diameter. From the incident and transmitted wave heights, wave reduction factor was computed.

## 3.5    Bulk drag coefficient of vegetation

Mazda et al. (1997a) estimated the effect of the flow resistance due to mangroves as a bottom friction. This drag coefficient, $C_d$, is approximated by:

$$C_d = \frac{32\sqrt{2}}{\pi} \frac{h^2}{H_{in}\Delta x} \left(\frac{H_{in}}{H_{trans}} - 1\right)$$

where, h is water depth, $H_{in}$ is the incident wave height, $H_{trans}$ is the transmitted wave height and $\Delta x$ is the distance between 2 sensors deployed in the field. $C_d$ is also influenced by the vegetation density. As waves travel over a vegetated bed, surface waves exert force on the plant stems, and in this process dissipate some of their energy (Mork, 1996).

The drag also depends on the flow conditions (Denny, 1988;Augustin et al., 2009). Two important numbers used to define the type of forces for given flow conditions are Reynolds number ($R_e$) and the Keulegan-Carpenter number (KC) (Keulegan and Carpenter, 1958). The Reynolds number is the ratio of inertial forces to viscous forces, and is useful to find out whether a flow is laminar or turbulent. It is given by Re = $U_w b/\nu$, where $U_w$ is horizontal oscillatory velocity scale, b is per unit blade length and $\nu$ is the kinematic viscosity of the fluid. The Keulegan-Carpenter number, KC represents the ratio of drag to inertial forces on a bluff body subject to oscillatory flow. The common form of KC is given as: KC = UT / B, where U is the velocity amplitude, T the period and B, the characteristic length scale (Keulegan and Carpenter, 1958). Previous studies have reported correlations between $C_d$ and non-dimensional quantities $R_e$ or KC (Mendez and Losada, 2004;Augustin et al., 2009; Bradley and Houser 2009; Paul and Amos, 2011). When $R_e$ is relatively small, the





flow is smooth and viscous forces dominate, and when Re is large, the flow is turbulent and inertial forces dominates. On the other hand, KC is relatively low when inertial forces dominate and high when drag forces dominate. Pinsky et al. (2013) reviewed all the earlier studies carried out in different habitats of vegetation at different locations, and estimated the value of $C_d$ based on the habitats (details related to only mangroves are listed in Table 1). The estimated average bulk drag coefficient from various field measurements for mangroves was 1.5 (Fig. 3). We have calculated $C_d$ using Mazda (1997b) equation, based on the measured data, and the value obtained was 0.5. The model was thus setup with the $C_d$ values obtained from both the methods; the results are discussed in the following sections.

## 4.    Results and Discussion

### 4.1    Analysis of measured data

The measured pressure data was analysed and wave characteristics were calculated using the zero-crossing method for each station using MATLAB programs developed by us. Wave statistics were calculated after de-trending the pressure for any low-frequency tidal component present. Significant wave heights and mean wave periods were extracted from the measured data. Significant wave heights (measured) and predicted tide elevations off Mumbai during 5 - 8 August 2015 are shown in Fig. 4.

Wind was relatively stable, and predominantly from the west-southwest direction near the coast during the above period; waves were approaching the coast nearly in the westerly direction. Due to logistics problem, measurements could be carried out only for one tidal cycle (in the night) each day. Water level was sufficient to make measurements in the vegetation area only on the first day, and in the subsequent days, water level was too low for taking measurements. The low wave heights recorded by the sensors are attributed to this reason.

### 4.2    Wave energy dissipation in the mangrove area

The tidal elevations were predicted using MIKE 21 inbuilt global tide model. Maximum water level predicted was 3.8m (Fig. 4). Maximum wave height of $\approx 0.3$m with mean wave period ranging between 3 and 6s was recorded only on the first day. Significant wave height ($H_s$) time series of each sensor (Fig. 5) show that wave heights experienced attenuation along the transect when the waves approached the vegetation zone. Reduction in wave height was the greatest (upto 52%) at P3 and the least (10%) at P2. The greatest wave height reduction was observed at P3 due to dense vegetation and attenuation of waves by the matrix of mangroves compared to that at P2. However, minimal change in mean wave period was observed when the waves travelled from P1 to P3 (Fig. 5), and wave periods ranged between 3 to 8s (except few higher values on 8 August 2015).

In the first two days, maximum $H_s$ observed at P1 location was 0.3 m and 0.28m at P2. During these days a very strong relation was observed (Fig. 6) between the water level and wave height (upto $R^2$=0.99). In the last 2 days of the measurements, waves with maximum $H_s$ of 0.18m at P1 and 0.15m at P2 were recorded, as the water level was relatively lower. At P3 location, the wave heights were small with maximum $H_s$ of $\approx 0.15$m on the first day of measurement period. It may be noted that location P2 lies inside the vegetation, and on the last day due to low water level, the corresponding wave heights were very small.    .





### 4.3 Model validation: No vegetation

Numerical experiments were conducted with various formulations in order to predict waves off Mumbai accurately. Initially, SWAN model was setup in standalone mode with default settings on the open boundaries (without boundary information from the WAVEWATCH 3 (WW3) model). The model results were validated with available wave data from the buoy deployed off Mumbai at 15m water depth during Oct - Nov 2009. The comparison showed underestimation in the modelled wave heights. The boundary conditions obtained from the (WW3) model was used to force the inner domain, and that resulted improvement in model results. Fig. 7 shows the comparison between modelled wave parameters with SWAN standalone and SWAN nested with WW3 and measured wave parameters. It is very evident from this comparison that nesting of SWAN with WW3 has captured swells arriving from as far as the Southern Ocean. It may be noted that the cyclone Phyan passed through the coastal area off Mumbai on 11 November 2009 (during this measurement period). However, ERA-I winds underestimated the cyclone winds, thereby predicting low $H_s$. As the study region was not under the direct influence of this cyclone, the maximum $H_s$ recorded ($\approx$2m) is comparatively lower than even the normal monsoon waves recorded in this region ($\approx$ 3-4m). Hence, this event was not given much attention for studying the wave attenuation due to extreme event. It is significant to note that other wave parameters(period and direction) showed considerable improvements when SWAN was nested within WW3 (Fig. 7).

### 4.4 Reduction in wave energy due to change in vegetation density and $C_d$

The vegetation parameters were varied in the numerical experiments to investigate model sensitivity to vegetation parameters. SWAN was run for a vegetation height of 3.0m with stem diameter varying between 0.1 and 0.3m and density of the mangroves between 0.175 and 0.350/m$^2$ (number of stems per m$^2$). To compute wave attenuation, the major parameter varied was drag coefficient $C_d$. The direction of the incident waves was taken as normal to the mangrove forest, as was the case when the measurements were performed. The vegetation was considered homogeneous with the characteristics in Table 2. It may be noted that the model was setup based on the bathymetry of ETOPO1 with 1km×1km resolution. This bathymetry data was augmented with the Naval Hydrographic Office (NHO) chart data for better resolution. Various sensitivity analyses were carried out with the vegetation module of SWAN to understand the role of different parameters affecting the wave attenuation process.

#### 4.4.1 Sensitivity analysis with vegetation

The transmitted wave heights were analysed under different groupings depending on the input parameters provided (vegetation density, vegetation diameter and drag coefficient). Wave attenuation through the mangrove forest was quantified using the wave reduction factor(r), defined by the following equation (Burger, 2005):

$$r = (H_{in} - H_{trans})/H_{in}$$

This factor could be linked directly to the effectiveness of the forest in attenuating waves. Wave reduction factor, from different cases, was compared with each other to understand relative importance of different vegetation parameters.

It was observed that wave attenuation increased with increase in $C_d$, density and stem diameter (Tables 3 and 4; Fig. 8 a&b). The resistance of the vegetation generates a drag force that causes reduction in wave



height (Mazda et al., 1997b). Model runs executed with $C_d$ values obtained from the literature ($C_d$ =1.5) and estimated for the Mumbai region ($C_d$ = 0.5) showed that attenuation varied from 55.69% to 49.93% (Table 3), that is, the change is ≈ 6%. When $C_d$ was further increased to 3.0, wave attenuation increased by about 10-15%.As shown in Table 3, wave attenuation has been computed with other $C_d$ values also. When the stem diameter was varied from 0.3m to 0.2m and 0.1m, wave attenuation decreased for any given $C_d$ values (Tables 3 and 4).

### 4.4.2 Wave height attenuation due to vegetation

SWAN was run with vegetation, and from the model runs, incident wave parameters and transmitted wave parameters were extracted at the vegetation area. With a vegetation density of $0.175/m^2$, stem diameter of 0.3m and drag coefficient varying from 0.4 to 1.5, the model reproduced attenuation, ranging from 49 to 55% (Table 3), which is comparable with the measurement (52%, refer to section 4.2). Vo-Loung and Massel (2008) studied attenuation in mangrove area in CanGio Mangrove Biosphere Reserve, Southern Vietnam, with number of trunks varying in the range of $1–21/m^2$ with mean diameter in the range 0.011–0.379m and found that reduction in wave height was about 20% over 100m in the mangrove forest. These numbers vary depending on the layers and the cells measured in mangrove site (Vo-Luong, 2006). Similarly, Narayan (2009) studied wave attenuation in Mangrove Island, considering the stem density varying between 0.5 and $1.7/m^2$ and vegetation width of 300m, and found that attenuation reached upto 60% at the port due to the effect of the mangrove island.

The present model results are in agreement with the above studies as well as the measurements carried out off Mumbai. However, the marginal difference found in the wave height reduction is due to vegetation parameters and resolution of the bathymetry considered in the model.

### 4.4.3 Wave spectral changes in the vegetation area

Time series measurements and model results support the hypothesis that the mangroves act as an efficient energy buffer in shallow and near-shore waters for a wide range of wind and wave conditions of typical meso- to macro-tidal coasts. Evidence for this role was found when the wave spectra obtained from model were compared. Typical 1D wave energy spectra were extracted at two locations, one in front of the vegetation (P1) and another inside the vegetation (P3). Fig. 9 shows an inter-comparison of wave energy spectra at both these locations for select time intervals. Wave energy was much less  at P3 than P1. These model results clearly indicate the contribution of mangrove vegetation as a friction factor to incoming waves.

### 5. Conclusions

The impact of coastal vegetation on wave attenuation has been investigated. The analysis of measured data collected from the mangrove forest off Mumbai presents wave attenuation, of the order of 50%, though width of the vegetation is not sufficient to provide greater wave attenuation. We find a linear relationship between wave height and water level in the nearshore region. A numerical wave model was set-up to study wave energy dissipation due to mangroves and sensitivity analyses were carried out with varying vegetation parameters. The numerical experiments show that for a vegetation density of $0.175/m^2$, stem diameter of 0.3m and drag coefficient varying from 0.4 to 1.5, the model reproduced wave attenuation, ranging from 49 to 55%, which is comparable with measurements (52%), and also with earlier studies. Fine resolution bathymetry will




enhance the accuracy of wave attenuation prediction in the shallow water areas covered with vegetation. The sensitivity analyses carried out for the mangrove forest off Mumbai provided knowledge on different vegetation parameters affecting the wave attenuation rate. This study has the potential of improving the quality of wave prediction in vegetation areas, especially during monsoon season and extreme weather events.

**Acknowledgements**

We thank Director, CSIR-NIO, Goa for providing facility to carry out this work. The first author acknowledges the Dept. of Sci& Tech, Govt. of India for supporting the research work through WOS-A(SR/WOS-A/ES-17/2012). The fieldwork data sharing is limited by our institute data sharing policy. The ERA-Interim wind data were downloaded from ECMWF (http://apps.ecmwf.int/datasets/). We are thankful to SWAN model developers for providing the source code. We acknowledge CSIR-NIO for providing high performance computing domain, HPC-Pravah for running the model. We are also thankful to Ravish Naik for his help during field work and Ankita Misra for helping in satellite image processing. The NIO contribution number is xxxx.

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

**Table 1. Review of $C_d$ calculated for mangroves (Ref. Pinsky et al., 2013).**

| Region | Study $C_d$ (estimated or assumed drag coeff) | Decay | $C_d$ (bulk drag coeff.) | Source |
|---|---|---|---|---|
| Australia | - | - | - | Massel et al. (1999) |
| Vietnam | 0.12 | 0.001 | 0.14 | Mazda et al. (1997a) |
| Vietnam | - | - | 2.72 | Mazda et al. (2006) |
| Vietnam | - | - | 2.69 | Quartel et al. (2007) |
| Vietnam | - | - | 0.42 | Vo-Luong and Massel (2006, 2008) |
| Mumbai, India | - | - | 0.50 | Present study |

**Table 2. Range of vegetation parameters considered for various model runs**

| Veg. height (m) | Stem diameter (m) | Density (no. of stems/m²) | $C_d$ |
|---|---|---|---|
| 3.0 | 0.3 - 0.1 | 0.175 - 0.350 | 0.2 - 3.0 |

**Table 3. Wave height attenuation (P1 to P3) (%) for different vegetation parameters (vegetation height = 3m; stem dia = 0.3m)**

| Density (no. of stems/m²) | $C_d$ | | | | | | | |
|---|---|---|---|---|---|---|---|---|
| | 0.2 | 0.4 | 0.5 | 0.6 | 0.8 | 1 | 1.5 | 3 |
| **0.175** | 47.25 | 49.14 | 49.93 | 50.76 | 52.08 | 53.27 | 55.69 | 60.6 |
| **0.2** | 47.55 | 49.72 | 50.53 | 51.33 | 52.79 | 54.03 | 56.54 | 61.58 |
| **0.25** | 48.17 | 50.51 | 51.52 | 52.37 | 54.01 | 55.26 | 58.02 | 63.28 |
| **0.3** | 48.61 | 51.31 | 52.43 | 53.44 | 55.04 | 56.63 | 59.4 | 64.68 |
| **0.35** | 49.14 | 52.08 | 53.27 | 54.31 | 56.07 | 57.51 | 60.6 | 65.96 |



**Table 4. Wave height attenuation (P1 to P3)(%) for different $C_d$ values and constant vegetation parameters (vegetation height = 3m; density = 0.175)**

| Stem dia (m) | $C_d$ | | | | | | | |
|---|---|---|---|---|---|---|---|---|
| | 0.2 | 0.4 | 0.5 | 0.6 | 0.8 | 1 | 1.5 | 3 |
| 0.1 | 45.6 | 46.38 | 46.82 | 47.25 | 47.84 | 48.54 | 49.93 | 53.27 |
| 0.2 | 46.38 | 47.84 | 48.54 | 49.14 | 50.2 | 51.14 | 53.27 | 57.51 |

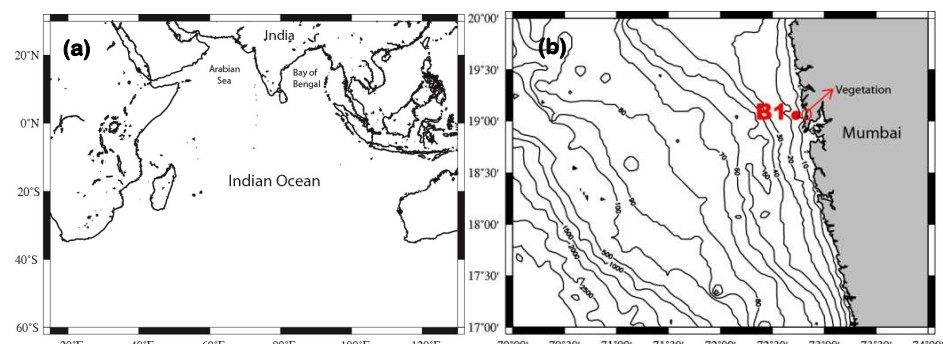

**Fig. 1. (a) Outer domain of the Indian Ocean chosen for modelling (b) Inner domain with depth contours off Mumbai, including buoy location**

20




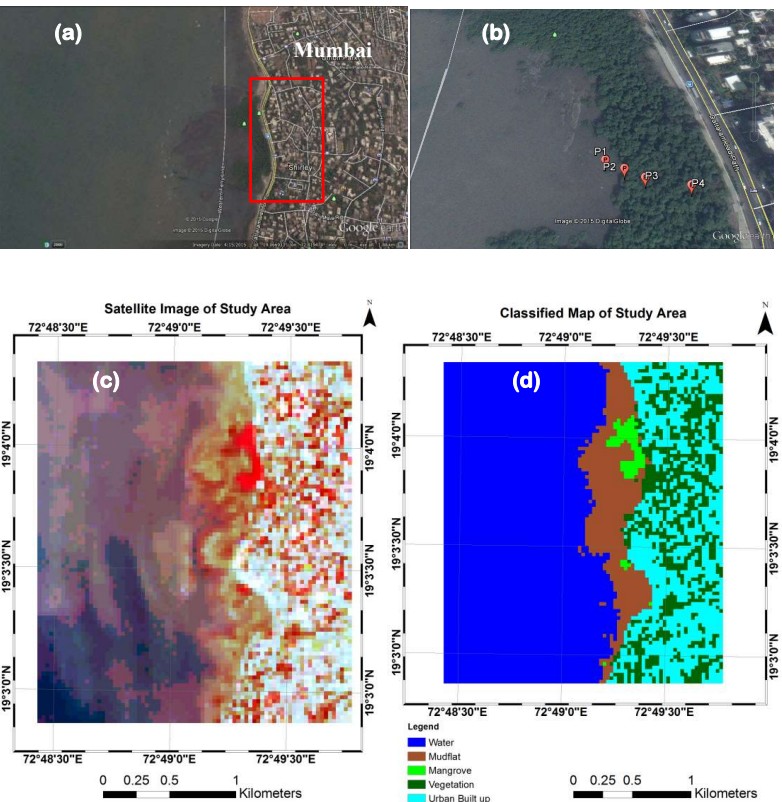

**Fig. 2 (a) Vegetation area off Mumbai (google image), (b) Domain showing the vegetation area and the**

5  **wave measurement locations (P1, P2, P3 and P4), (c) satellite image and (d) classified image of the study**

**area.**

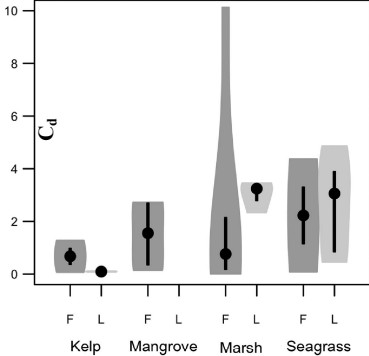

**Fig. 3.Violin plot of drag coefficients ($C_d$) across kelp, mangrove, marsh and seagrass habitats from lab**

10  **(L) and field (F) studies. Width of polygon indicates the kernel density, dot marks the median and thick**

**black bar marks the inter-quartile range *(Ref. Pinsky et al., 2013)*.**





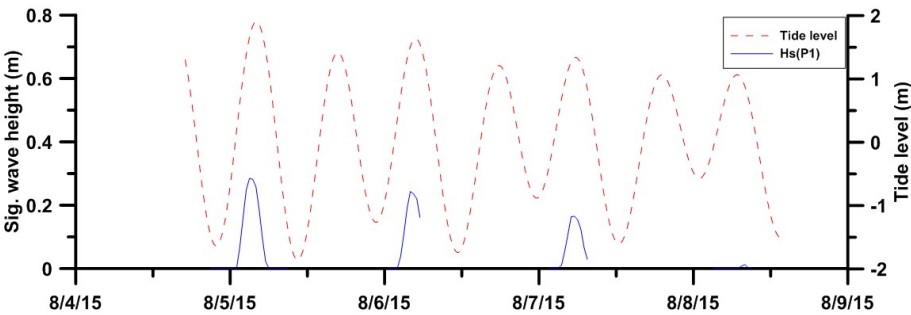

**Fig. 4. Significant wave heights and predicted tide elevations off Mumbai during 5-8 August 2015.**

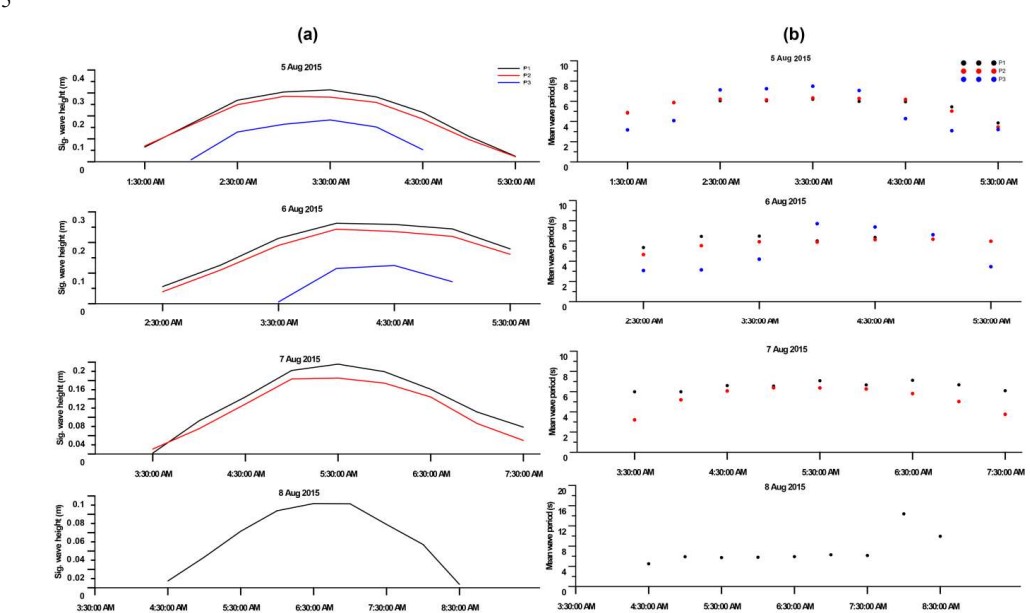

**Fig. 5. (a) Significant wave height, $H_s$ and (b) mean wave period, $T_m$ at the 4 measurement locations (in front of and inside the vegetation) during 5-8 August 2015.**



**Fig. 6. Linear relation between water level and sig. wave height in the study region on 5$^{th}$ August 2015 (left) and 6$^{th}$ August 2015 (right) at the locations P1(a, d), P2 (b,e) and P3(c,f).**

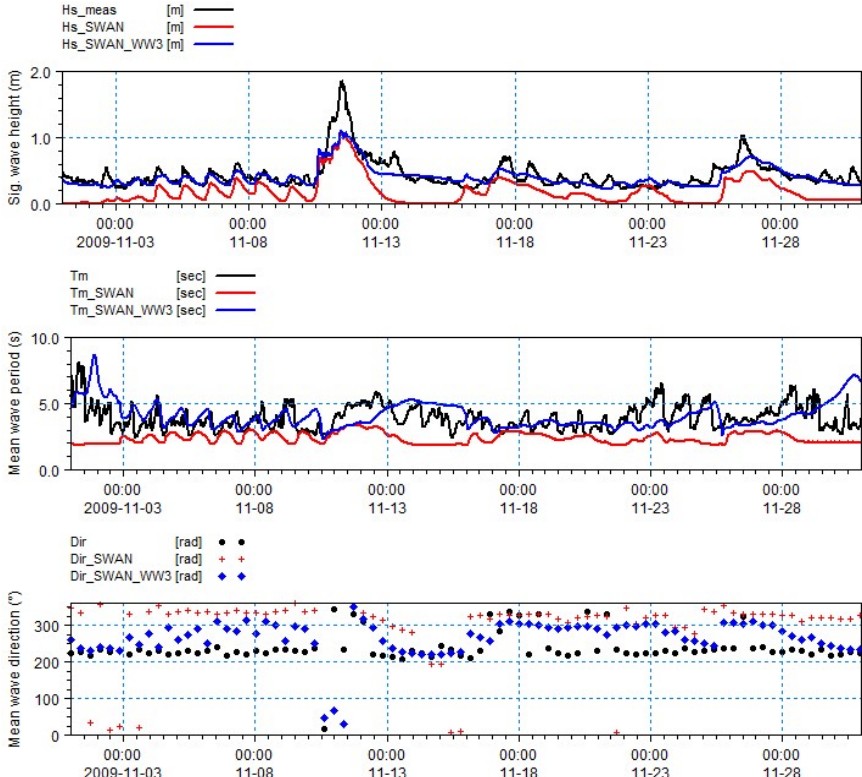



**Fig. 7. Comparison of SWAN wave model results with buoy data (without vegetation)**

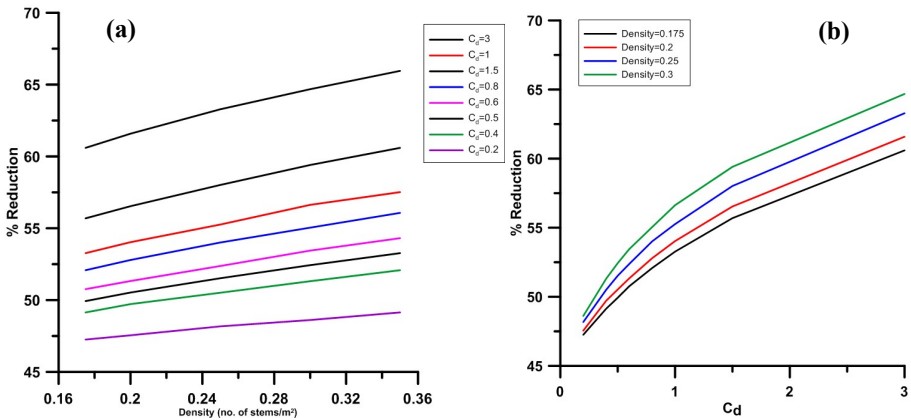

5    **Fig. 8. Wave height attenuation: (a) varying vegetation density and different $C_d$ and**

   **(b) varying $C_d$ for different vegetation density**

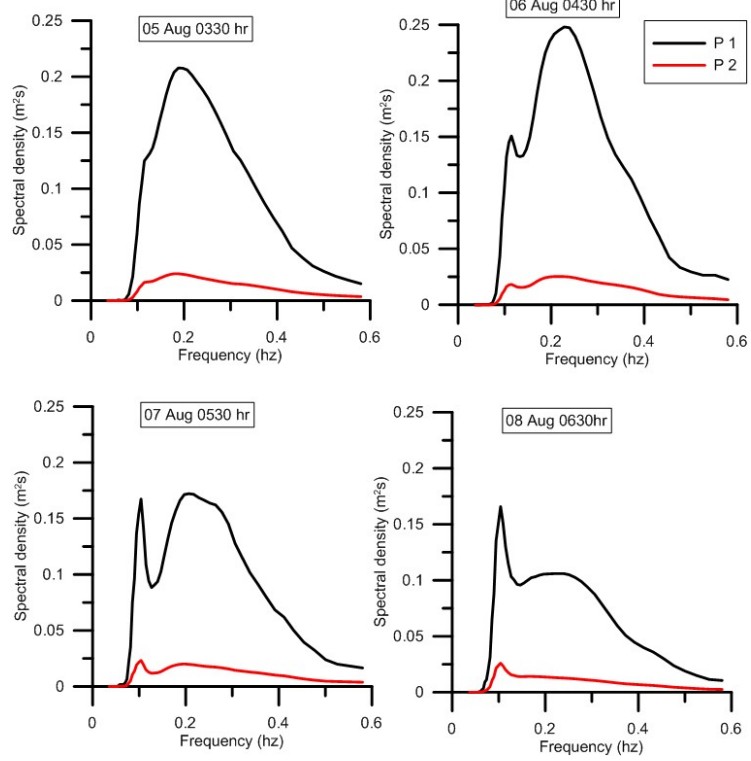

**Fig. 9. Wave spectral comparison at two locations P1 (in front of vegetation) and P3 (inside the**

10    **vegetation) at select time intervals in August 2015 obtained from SWAN.**