# Peer review of "Wave energy dissipation in the mangrove vegetation off Mumbai, India"

_Ocean Science, 2018_

## Referee Comment (RC1) · L. Bricheno (Referee) · 5 Jul 2018

Volvaiker (Ocean sciences)

Summary

This manuscript presents modelling and observations of how mangroves dissipate incoming wave energy. Authors use a combination of local wave modelling and pressure level sensors are combined to study the dissipation of energy, when surface waves approaching the coast pass through an area of vegetation. The study is focussed on an area of mangrove forest in Mumbai. The authors discuss how waves are affected by interactions with different kinds of vegetation / parts of the same plant. A sensitivity study is performed of how characteristics of plants affect wave attenuation. Different

methods of calculation for Cd are also tried.

Overview This is a well written paper on a topic and geographical area which has not been well studied. There is limited observation data, over a very short window. It could be strengthened by looking at different periods, otherwise the statements about extreme events are just speculation.

Major issues

During the short window of observations, the waves were very low. Therefore we cannot draw conclusions about the dissipation of extreme wave conditions. Also, the tidal range meant that waves are only present for a fraction of the observation period. (I suppose this is obvious as the mangroves habitat is inter-tidal) but a longer period of observations, with variable wave conditions might have benefited the study. Would you like to speculate on how extreme events like cyclones might differ from the results you have shown here?

In the conclusion you state that "this study has potential of improving the quality of wave prediction in vegetation areas, especially during monsoon season and extreme weather events". I think this a very strong statement to end on, because you have so far only shown skill in representing low waves, and their propagation through mangroves. I don't think this paper has yet demonstrated the effectiveness of the model at high wave / deep inundation which would occur during monsoon conditions. Suggest if you wanted to add a section speculating on this, it would be nice to see, but you would need to observe, or at least model an extreme event with high waves. It would also be interesting to see how well this model behaves during a large storm- surge, when the waves are approaching the mangrove on a higher background water level. I expect that this would change which parts of the plants are submerged, and thus the drag effects calculated.

Minor corrections

P2 L36-38 focus of the study - suggest moving this line to section 2.

use of hectares (P4 L17 - can we have SI unit, or square km instead?)

P4: please state the water depth at each observation site P1 (offshore) to P4 inshore. Maybe give the min/max water depth over a tidal / spring-neap cycle?

in section 3.3 please explain first that the model you are using to focus on the mangroves is SWAN. Then separate out the description of the WaveWatch Indian Ocean model. Reading it right now, it looks at first reading as if the IO model is also SWAN.

Add a reference for the ERA-I winds (Dee et al. 2011)

P5 L8 "ERA-I winds..." repetition. cut this, and move the spatial resolution to L3/4 above.

P7 MIKE 21 inbuilt global tide model. What is this? Can you provide a reference?

P8 L4 "The model results..." suggest change to "The SWAN only model results..." as we were just taking about WW3. This section is talking about the stand-alone SWAN, without swells from the WW3 model.

P8 L20 repetition of "vegetation parameters" suggest changing the end of sentence to "sensitivity to presence of vegetation"

P9 L28-30, Explain that Fig 9. shows a comparison of consecutive days, at similar phases of the tide. This is important as it makes the results more comparable. Also important to note that the spectral shape, and wave period remain the same, and just wave heights are attenuated. Could even plot the spectra in Fig. 9 on 2 different vertical axes (with peaks scaled to be equal), so we can examine the shape more closely?

Acknowledgements the NIO contribution number is missing

Some typographic errors, mostly with whitespaces missing around brackets. E.g. P3 L22 "..coast atleast during.." P4 L2. P4 L25. P5 L6. P7 L28 (upto 52%) -> (up to 52%)

P8 L33. P9 L4. P10 L8

Extra reference Dee, D., S. Uppala, A. Simmons, P. Berrisford, P. Poli, S. Kobayashi, U. Andrae, M. Balmaseda, G. Balsamo, P. Bauer, and P. Bechtold (2011), The ERA-Interim reanalysis: Configuration and performance of the data assimilation system., Quart. J. Royal Met. Soc., 137, 553–597.

problems with formatting refs. P11 L23 and L25 Maxda Missing ref. to MIKE tidal model

Swap ordering of figures, so they appear relative to where they are mentioned in the text. Swap Fig 1 and 2.

Figure 1: this is a very large nesting ratio, going from 0.5 degree resolution Indian Ocean WW3 to 1 minute SWAN model. Is this likely to be a problem? Why was the larger IO domain chosen, would it be better to have an intermediate step?

Figure 2: How does the algorithm distinguish between mangroves and other vegetation?

Figure 4: clarify caption. Is this showing observations of SWH and MIKE modelled tidal water levels? Ditto figure 5: are these showing observations only, not model? Please add both points and lines to both plots. Also, in Fig 5. Is it 3 measurement sites, not 4?

Figure 6: strong linear relationship, somewhat undermined in e.g. panels (c) and (f) because of zero waves skewing the fit. maybe remove these points? If there is zero wave height, but still positive water depth, then what is happening at these times in P3?

Figure 7. Clarify caption to read "SWAN wave height attenuation..."
* * *

---

## Referee Comment (RC2) · H Karunarathna (Referee) · 7 Aug 2018

The manuscript presents numerical study of wave attenuation by mangroves in a site in India and some field investigations to support wave modelling. The impacts of plant stem diameter, plant density and drag coefficient are discussed. The paper is well written in general. The abstract reflects the contents of the paper. Figures are reasonably clear and informative. My major concerns regarding the research presented in this paper are: 1. Wave conditions used for modelling are very mild and do not in any way represent the conditions during which wave attenuation will be important. Even though the authors mention that wave attenuation is important during tsunamis or high energy wave events such as tropical cycles, wave conditions used in this study do not reflect such conditions. Therefore, the significance of this study is questionable.

[Figure]

2. Mangroves have a very complex root/branch system. The application of a very simple model to represent mangroves may have serious implications on the credibility of results. Those simple models are used in previous studies to represent plants like seagrass where the structure of the plant is relatively simple. 3. Storm surge during an extreme event can be very significant for determining water levels and wave propagation in mangrove forests. It is therefore, necessary to investigate high energy wave conditions with surge before concluding wave attenuation capacity of mangroves and the ability of the numerical model to capture such phenomena.

---

## Author Comment (AC1) · 24 Aug 2018

**Response to Reviewer 1:**

Summary
This manuscript presents modelling and observations of how mangroves dissipate incomingwave energy. Authors use a combination of local wave modelling and pressurelevel sensors are combined to study the dissipation of energy, when surface wavesapproaching the coast pass through an area of vegetation. The study is focussed onan area of mangrove forest in Mumbai. The authors discuss how waves are affectedby interactions with different kinds of vegetation / parts of the same plant. A sensitivitystudy is performed of how characteristics of plants affect wave attenuation. Differentmethods of calculation for Cd are also tried.

Overview This is a well written paper on a topic and geographical area which has notbeen well studied. There is limited observation data, over a very short window. Itcould be strengthened by looking at different periods, otherwise the statements aboutextreme events are just speculation.

**Thanks to the reviewer for her good words about our work, and for her 2 major specific comments to the article. Also, appreciate the reviewer for going through the MS meticulously. All corrections suggested are incorporated in the revised MS. The authors agree with the fact that the duration of data collection period was limited due to several constraints. The reviewer correctly pointed out that the statement about extreme events is just a speculation, and therefore this statement is removedfrom the manuscript for clarity to the readers.**

Major issues
During the short window of observations, the waves were very low. Therefore wecannot draw conclusions about the dissipation of extreme wave conditions. Also, thetidal range meant that waves are only present for a fraction of the observation period.(I suppose this is obvious as the mangroves habitat is inter-tidal) but a longer period ofobservations, with variable wave conditions might have benefited the study. Would youlike to speculate on how extreme events like cyclones might differ from the results youhave shown here?

**Thanks to the reviewer for this specific comment. It is true that this study doesnot show directly any attenuation during extreme weather event. As stated in the manuscript, this is a preliminary study on wave attenuation characteristics in the vegetation zone along the Indian coast using measurement and modelling during high wave energy conditions (monsoon season). The authors fully agree with the reviewer that during the short time window of observations (August 2015), the wave activity was very low. Also, due to many constraints and logistics issue, the field data collection was limited to a short time window only. The authors agree with the viewpoint that longer period of observations with variable wave conditions would have provided better results on the wave dissipation characteristics by mangroves. We admit that this is a limitation of the present study. A more detailed and rigorous exercise with planned field campaigns iswarranted in a better perspectiveto understand the dissipative effects due to mangroves, and that forms the scope of future work. This sentence is now added in the 'Conclusions' section of the revised manuscript.**

In the conclusion you state that "this study has potential of improving the quality of wave prediction in vegetation areas, especially during monsoon season and extreme weather events". I think this a very strong statement to end on, because you have so far only shown skill in representing low waves, and their propagation through mangroves. I don't think this paper has yet demonstrated the effectiveness of the model at high wave / deep inundation which would occur during monsoon conditions. Suggest if you wanted to add a section speculating on this, it would be nice to see, but you would need to observe, or at least model an extreme event with high waves. It would also be interesting to see how well this model behaves during a large storm-surge, when the waves are approaching the mangrove on a higher background water level. I expect that this would change which parts of the plants are submerged, and thus the drag effects calculated.

**Thank you for making a specific comment and suggestion. In a climatological perspective and based on the best track record of cyclones and severe cyclones that form over the north Indian Ocean region, the frequency of cyclones in the Arabian Sea is much less as compared to that in the Bay of Bengal. As mentioned earlier, we have already removed the statement on wave energy dissipation during extreme weather events. However, we shall surely include this good suggestion in our planned future study.**

Minor corrections
P2 L36-38 focus of the study - suggest moving this line to section 2. use of hectares (P4 L17 - can we have SI unit, or square km instead?)

**As suggested, P2L36-38 is now moved to Section 2, and the use of hectares (P4 L17) is converted to square km in the revised manuscript.**

P4: please state the water depth at each observation site P1 (offshore) to P4 inshore. Maybe give the min/max water depth over a tidal / spring-neap cycle? in section 3.3 please explain first that the model you are using to focus on the mangroves is SWAN. Then separate out the description of the WaveWatch Indian Ocean model. Reading it right now, it looks at first reading as if the IO model is also SWAN. Add a reference for the ERA-I winds (Dee et al. 2011).

**As suggested by the reviewer, water depth details are given as below:**
**The measured maximum water depths at each sensor were 2m (P1), 1.5m (P2), 1m (P3) and 0.3m (P4) during high tide; it may be noted that during low tide, these locations were exposed.**

**Section 3.3 has been modified as follows:**
**The third generation numerical model SWAN (Simulating Waves Nearshore) was specifically developed for finite water depth applications (Booij et al., 1999). The governing equation in the model is the wave action balance equation with various source and sink terms. The bathymetry was generated with ETOPO1 Earth Topography (1 minute) data obtained from the National Geophysical Data Centre, USA. The SWAN domain (17°N to 20°N and 70°E to 74°E) was set with a spatial resolution of 0.01°×0.01° (Fig. 1b). As the size of actual patch of the vegetation is approximately of one grid size, one single grid was**

considered with vegetation. ERA-I winds (Dee et al. 2011) with the resolution of 0.125°×0.125° were used as input.

The model discretization considered 31 frequency bins ranging from 0.05 to 1.00 Hz on a logarithmic scale, and 36 directional bins with an angular resolution of 10°. The SWAN setup in the present study used Cavaleri and Malanotte-Rizzoli (1981) wave growth physics, and shallow water triad non-linear interaction using the lumped triad approximation of Eldeberky (1996). The model was initiated with modified white-capping dissipation (Jansen, 1991) which is the default formulation in SWAN model. The quadruplet non-linear wave-wave interaction was computed using the Discrete Interaction Approximation theory (Hasselman et al., 1985). The depth induced breaking was computed using spectral version of the model with breaking index, $\gamma = 0.73$ (Battjes and Janssen, 1978). The bottom friction in SWAN was calculated based on the Collins formulation (Collins, 1972) with a friction coefficient, $c_{fw} = 0.02$ $m^2s^{-3}$. The model was also run with different bottom friction physics such as MADSEN and JONSWAP available in the model. However, we found that the results are better with Collins formulation. Therefore, all model runs in this study were simulated using Collins bottom friction. The boundary files containing 2D directional wave spectra were generated along the SWAN model domain using WAVEWATCH III (WW3) model with a spatial resolution of 0.5°×0.5°. The WW3 domain covers the entire Indian Ocean from 60°S to 30°N and 15°E to 130°E (Fig. 1a), and accommodates the distant swells propagating from the South Indian Ocean/Atlantic Ocean into the North Indian Ocean (Aboobacker et al., 2011; Samiksha et al., 2012; Sabique et al., 2012).

P5 L8 "ERA-I winds..." repetition. cut this, and move the spatial resolution to L3/4above.

Modified as suggested by the reviewer.

P7 MIKE 21 inbuilt global tide model. What is this? Can you provide a reference?

MIKE 21 is a depth averaged two-dimensional hydrodynamic model, MIKE 21 HD, developed by DHI Water & Environment, Denmark (http://www.dhigroup.com). It has an inbuilt Global Tide Model data which represents the major diurnal (K1, O1, P1 and Q1) and semidiurnal tidal constituents (M2, S2, N2 and K2) with a spatial resolution of 0.25° × 0.25° based on TOPEX/POSEIDON altimetry data.

Reference is added as below:
DHI, MIKE 21 toolbox, Global tidal model, in: Scientific Documentation, 2014, pp142

P8 L4 "The model results..." suggest change to "The SWAN only model results..." aswe were just taking about WW3. This section is talking about the stand-alone SWAN,without swells from the WW3 model.

**Modified as suggested by the reviewer.**

P8 L20 repetition of "vegetation parameters" suggest changing the end of sentence to"sensitivity to presence of vegetation"

**As suggested, necessary corrections are made in the revised manuscript.**

P9 L28-30, Explain that Fig 9. shows a comparison of consecutive days, at similarphases of the tide. This is important as it makes the results more comparable. Alsoimportant to note that the spectral shape, and wave period remain the same, and justwave heights are attenuated. Could even plot the spectra in Fig. 9 on 2 different verticalaxes (with peaks scaled to be equal), so we can examine the shape more closely?Acknowledgements the NIO contribution number is missing

**We have included the first comment of the reviewer in the revised MS as follows:**
**Fig. 9 shows an inter-comparison of wave energy spectra at both the locations for select time intervals during consecutive days (high tide).**

**We are so sorry that we are unable to do the second correction as suggested in Fig.9.**
**NIO contribution number will be added in the final version.**

Some typographic errors, mostly with whitespaces missing around brackets. E.g. P3L22 "..coastatleast during.." P4 L2. P4 L25. P5 L6. P7 L28 (upto 52%) -> (up to 52%)P8 L33. P9 L4. P10 L8

**Corrected as suggested by the reviewer.**

Extra reference Dee, D., S. Uppala, A. Simmons, P. Berrisford, P. Poli, S. Kobayashi,U. Andrae, M. Balmaseda, G. Balsamo, P. Bauer, and P. Bechtold (2011), The ERAInterimreanalysis: Configuration and performance of the data assimilation system.,Quart. J. Royal Met. Soc., 137, 553–597.

**Reference is added**

problems with formatting refs. P11 L23 and L25 Maxda Missing ref. to MIKE tidalmodel

**Modified as suggested by the reviewer.**

Swap ordering of figures, so they appear relative to where they are mentioned in thetext. Swap Fig 1 and 2.

**Fig.1 explains the study area and the model domains and it is placed first in the text. This is followed by Fig.2, which explains the vegetation details off Mumbai.**

Figure 1: this is a very large nesting ratio, going from 0.5 degree resolution IndianOcean WW3 to 1 minute SWAN model. Is this likely to be a problem? Why was thelarger IO domain chosen, would it be better to have an intermediate step?

**Thanks for the query. As stated in the manuscript the waves in the north Indian Ocean are affected by the distant swells propagating from the South Indian Ocean/Atlantic Ocean. Our earlier studies (Aboobacker et al., 2011; Samiksha et al., 2012; Sabique et al., 2012) have proved the impact of these distant swells on the wave climate in the north Indian Ocean during different seasons. Therefore, in the present study, we have considered the entire Indian Ocean from 60°S to 30°N and 15°E to 130°E (Fig. 1a) in the model domain.**

**Also, the main aim of nesting the SWAN model with WW3 model was to accommodate the distant swells, and for this we have generated, 2D spectra files at the boundaries of the SWAN model domain; there is no technical issue (CFL criteria) due to nesting ratio.**

Figure 2: How does the algorithm distinguish between mangroves and other vegetation?

**Thanks for the query. Accordingly, Section 3.1 is thoroughly modified as follows:**

**Landsat5 TM (9 January 2015) satellite dataset (Fig. 2c), obtained from the global land cover facility site with a resolution of 30 m, has been used to estimate the distribution of mangroves off Carter Road. This area has been classified based on the Iterative Self-Organizing Data Analysis Technique (ISODATA) algorithm (Memarsadeghi et al, 2007). An unsupervised classification method, ISODATA classifies pixels into spectral clusters based on similar spectral characteristics in the input band. A minimum distance criterion is then used to assign each pixel to the "nearest" cluster. For this study, 5 classes are extracted- water, mudflat, mangrove, vegetation and urban (Fig. 2d) using the ERDAS 9.1 unsupervised classification tool, and ARC GIS 10.1 is used to make the classification Map. An accuracy assessment is further carried out using GCPs collected during field measurements and the overall accuracy obtained for the classification is 93.5%. Finally, as the focus of this study is confined to mangrove region, the area covering mangroves is calculated, separating out the vegetation, and it is estimated to be about 0.08 km$^2$.**

Figure 4: clarify caption. Is this showing observations of SWH and MIKE modelled tidalwater levels? Ditto figure 5: are these showing observations only, not model? Pleaseadd both points and lines to both plots. Also, in Fig 5. Is it 3 measurement sites, not 4?

**Figure 4: Yes. It is showing observed significant wave height and predicted tide elevations. Accordingly, the caption is modified as below:**

**"Fig. 4. Observed significant wave heights and predicted tide elevations (MIKE Global Tide Model) off Mumbai during 5-8 August 2015."**

**Figure 5: Yes, it is showing all observations measured at 3 locations (due to low water level no measurements were recorded at location P4).**

**Modified Figure 5 is given below:**

[Figure]

Figure 6: strong linear relationship, somewhat undermined in e.g. panels (c) and (f)because of zero waves skewing the fit. maybe remove these points? If there is zerowave height, but still positive water depth, then what is happening at these times in P3?

**We thank the reviewer for this good comment. Accordingly, we have removed the points with zero wave heights, and replotted the figure (given below).**

**We assume that the reviewer meant P4 (not P3). Due to very low water level, no measurements could be recorded at location P4.**

**Modified Figure 6:**

[Figure]

Figure 7. Clarify caption to read "SWAN wave height attenuation..."

**The plot shows only the validation of SWAN model results with the measured data (without vegetation). No wave height attenuation is depicted in the plot.**

---

## Author Comment (AC2) · 24 Aug 2018

**Response to Reviewer 2:**

The manuscript presents numerical study of wave attenuation by mangroves in a sitein India and some field investigations to support wave modelling. The impacts of plantstem diameter, plant density and drag coefficient are discussed. The paper is wellwritten in general. The abstract reflects the contents of the paper. Figures are reasonablyclear and informative.

**Thanks to the reviewer for her good words about our work, and also for the 3 major specific comments to the article. The authors appreciate the comments and constructive criticism of the reviewer, and that helped in improving the manuscript. All suggested corrections are incorporated in the revised MS.**

My major concerns regarding the research presentedin this paper are:

1. Wave conditions used for modelling are very mild and do notin any way represent the conditions during which wave attenuation will be important.Even though the authors mention that wave attenuation is important during tsunamisor high energy wave events such as tropical cycles, wave conditions used in this studydo not reflect such conditions. Therefore, the significance of this study is questionable.

**Thanks to the reviewer for this specific comment. It is true that this study does not show directly any attenuation during extreme weather event. As stated in the manuscript, this is only a preliminary study on wave attenuation characteristics in a vegetation zone along the Indian coast using measurement and modelling, of course, during high wave energy conditions (monsoon season) in the open ocean. The authors fully agree with the reviewer that during the short span of observations (August 2015), the wave activity was very low. Also, due to many constraints and logistics issue, the field data collection was limited to a short time window only; longer period of observations with variable wave conditions would have provided better results on the wave dissipation characteristics by mangroves. We admit that this is a limitation of the present study. A more detailed and rigorous exercise with planned field campaigns is warranted in a better perspective to understand the dissipative effects due to mangroves, and that forms the scope of future work. This sentence is now added in the 'Conclusions' section of the revised manuscript. We have already removed the statement on wave energy dissipation during extreme weather events from the manuscript.**

**It may be noted that both the reviewers have given the same comments. Hence, our response is more or less, the same.**

2. Mangroves have a very complex root/branch system. The application of a verysimple model to represent mangroves may have serious implications on the credibilityof results. Those simple models are used in previous studies to represent plants likeseagrass where the structure of the plant is relatively simple.

We thank the reviewer for the comment. The best available numerical description of the effect of vegetation on waves is based on the representation of vegetation by vertical, ridged cylinders that was derived by Dalrymple et al. (1984). To account for wave dissipative effects due to vegetation in SWAN, the model includes wave damping over a vegetation field at variable depths using the cylinder approach suggested by Dalrymple et al. (1984) and modified by Mendez and Losada (2004) and The SWAN Team (2015). In this method, the vegetation is modeled as cylindrical obstacles causing a drag force and translated into an amount of energy that gives an energy dissipation term. The physical mechanism behind wave dissipation by vegetation is detailed in the following work (Dalrymple et al., 1984; Kobayashi et al., 1993; Mendez and Losada, 2004; Suzuki et al., 2011). Suzuki et al. (2011) further modified the SWAN model to include a vertical layer schematization for the mangrove vegetation. As model results have been extensively validated in these studies against field studies and experimental observations in the past, this model is confidently used in the present study.

3. Storm surge duringan extreme event can be very significant for determining water levels and wave propagationin mangrove forests. It is therefore, necessary to investigate high energy waveconditions with surge before concluding wave attenuation capacity of mangroves andthe ability of the numerical model to capture such phenomena.

The authors appreciate the reviewer for his constructive comment. High energetic waves and storm surges resulting from extreme weather events such as tropical cyclones are not considered in this study. The authors fully agree with the comment that during extreme events like tropical cyclones, the total water levels in the nearshore is resultedfrom a combined effect of reduced atmospheric pressure, storm surge and wave-induced setup. As stated in above reply to Query 1, this is a limitation of the present study. A more detailed and rigorous exercise with planned field campaigns is warranted in a better perspective to understand the dissipative effects of mangroves during extreme weather events, and that forms the scope of future work. This sentence is now added in the 'Conclusions' section of the revised manuscript.